# Trifluoperazine and Its Analog Suppressed the Tumorigenicity of Non-Small Cell Lung Cancer Cell; Applicability of Antipsychotic Drugs to Lung Cancer Treatment

**DOI:** 10.3390/biomedicines10051046

**Published:** 2022-04-30

**Authors:** Joo Yeon Jeong, Haangik Park, Hong Yoo, Eun-Jin Kim, Borami Jeon, Jong Deog Lee, Dawon Kang, Changjoon Justin Lee, Sun Ha Paek, Eun Joo Roh, Gwan-Su Yi, Sang Soo Kang

**Affiliations:** 1Department of Anatomy & Convergence Medical Science, Institute of Health Sciences, College of Medicine, Gyeongsang National University, Jinju 52727, Korea; jooya@gnu.ac.kr; 2Department of Bio and Brain Engineering, Korea Advanced Institute of Science and Technology, Daejeon 34141, Korea; winggpark@kaist.ac.kr; 3Division of Pulmonology and Allergy, Department of Internal Medicine, Gyeongsang National University Hospital, College of Medicine, Gyeongsang National University, Jinju 52727, Korea; ryu2030@gnu.ac.kr (H.Y.); ljd@gnu.ac.kr (J.D.L.); 4Department of Physiology & Convergence Medical Science, Institute of Health Sciences, College of Medicine, Gyeongsang National University, Jinju 52727, Korea; eunjin@gnu.ac.kr (E.-J.K.); dawon@gnu.ac.kr (D.K.); 5Chemical Kinomics Research Center, Korea Institute of Science and Technology, Seoul 02792, Korea; 112511@kist.re.kr (B.J.); r8636@kist.re.kr (E.J.R.); 6Center for Glia-Neuron Interaction and Neuroscience, Korea Institute of Science and Technology, Seoul 02792, Korea; cjl@kist.re.kr; 7Center for Cognition and Sociality, Institute for Basic Science, Daejeon 34141, Korea; 8Department of Neurosurgery, College of Medicine, Seoul National University, Seoul 03080, Korea; paeksh@snu.ac.kr

**Keywords:** non-small cell lung cancer, trifluoperazine, antipsychotics, apoptosis, proliferation

## Abstract

Despite significant advances in diagnostic and therapeutic technologies, lung cancer remains the leading cause of cancer-related mortality worldwide. Non-small cell lung cancer (NSCLC) accounts for approximately 85% of lung cancer cases. Recently, some antipsychotics have been shown to possess anticancer activity. However, the effects of antipsychotics on NSCLC need to be further explored. We examined the effects of trifluoperazine (TFP), a commonly used antipsychotic drug, and its synthetic analogs on A549 human lung cancer cells. In addition, cell proliferation analysis, colony formation assay, flow cytometry, western blot analysis, and in vivo xenograft experiments were performed. Key genes and mechanisms possibly affected by TFP are significantly related to better survival outcomes in lung cancer patients. Treatment with TFP and a selected TFP analog 3dc significantly inhibited the proliferation, anchorage-dependent/independent colony formation, and migration of A549 cells. Treatment with 3dc affected the expression of genes related to the apoptosis and survival of A549 cells. Treatment with 3dc promoted apoptosis and DNA fragmentation. In all experiments, including in vivo studies of metastatic lung cancer development, 3dc had more substantial anticancer effects than TFP. According to our analysis of publicly available clinical data and in vitro and in vivo experiments, we suggest that some kinds of antipsychotics prevent the progression of NSCLC. Furthermore, this study indicates a synthetic TFP analog that could be a potential therapeutic for lung cancer.

## 1. Introduction

Lung cancer is the most common cause of cancer-related mortality in the world [1]. Treatment against lung cancer can involve the surgical removal of cancer, chemotherapy, radiation therapy, or a combination of these treatments. Treatment decisions for a given individual must account for the location and extent of the tumor and the overall health status of the patient. Notably, non–small cell lung cancer (NSCLC) patients account for 80% of all patients with the most common type of lung cancer. When patients are diagnosed with NSCLC, usually the status of the disease is inoperable and they need systemic therapy. However, resistance to chemotherapy or targeted therapies such as epidermal growth factor receptor–tyrosine kinase inhibitors is a major problem in systemic lung cancer treatment [2]. Thus, there is an urgent need to identify therapeutics that overcome drug resistance.

Recently, antipsychotics, antidepressants, and neuroleptics have shown remarkable antiproliferative activity [3]. Moreover, a low prevalence of various cancer types was observed in patients with schizophrenia [4,5]. In the case of lung cancer, several cohort studies have reported cases of reduced incidence ratio in the schizophrenic population [6,7,8,9,10,11,12,13]. This implies that some factors of schizophrenic patients including neuroleptic treatment might have an anticancer effect. However, the patient-level analysis related to the treatment of specific antipsychotics is not sufficiently investigated. Based on these biological and epidemiological grounds, it is necessary to further analyze the anticancer effect of antipsychotics.

Trifluoperazine (TFP) is a phenothiazine derivative and is generally used as an antipsychotic agent due to its blocking effect of the dopamine receptor [14]. Interestingly, TFP is also used as a calmodulin (CaM) antagonist [15]. CaM antagonists were reported to suppress the proliferation of cancer cells [3,16,17] and induce apoptotic cell death [18]. In addition, TFP boosts the cytotoxicity of the radiomimetic agent bleomycin [19].

TFP is primarily used as a schizophrenia treatment. However, its use has declined in many parts of the world because of its association with highly frequent and severe early and late tardive dyskinesia, a type of extrapyramidal syndrome. All antipsychotics can cause this rare and sometimes fatal neuroleptic malignant syndrome [20]. The present study aimed to evaluate the possibility of antipsychotics as anticancer therapeutics for NSCLC. In addition, the potential anti-lung-cancer activity of TFP and its analogs was examined using in vitro and in vivo experiments.

## 2. Materials and Methods

### 2.1. Materials

As previously reported [21], TFP analogs (3db, 3dc, 3dd, and 3fb) were synthesized by Dr. Roh (KIST) and dissolved in distilled water (DW). Stock solutions were stored at −20 °C and diluted in a culture medium before each treatment. TFP was purchased from Sigma-Aldrich (St. Louis, MO, USA) and dissolved in DW immediately before use.

### 2.2. Microarray Data Processing 

We downloaded three lung cancer patient transcriptome microarray datasets including data for survival time (GSE19188 [22], GSE31210 [23], and GSE50081 [24]) from the National Center for Biotechnology Information Gene Expression Omnibus (NCBI GEO, https://www.ncbi.nlm.nih.gov/geo/) database (accessed on 22 March 2022) [25]. Every dataset was obtained from the same platform, the Affymetrix Human Genome U133 Plus 2.0 Array (GPL570). Raw data CEL files of each dataset were downloaded from the GEO supplementary files repository and processed by the robust multiarray average (RMA) algorithm [26]. The expression values within each dataset were normalized by log2-transformation and quantile normalization. Probes in the microarray datasets were translated to Entrez gene IDs according to the platform annotation data table. Probes that were matched to identical gene IDs were averaged. If multiple gene IDs were assigned to a single probe ID, each gene ID was separated, and the expression value of the original probe was used. We conducted integrative analysis to reduce data heterogeneity, including batch effects, and enhance the analysis’s statistical power. Every dataset was combined using ComBat [27], a batch effect adjustment method based on an empirical Bayes model. In addition, ComBat with the parametric adjustment option was applied to preprocessed datasets.

### 2.3. Curation and Scoring of Functional Gene Set

To collect biologically and functionally related gene sets which contain genes of interest, we downloaded canonical pathway (CP) and gene ontology (GO) terms [28] and their corresponding gene lists from the Molecular Signatures Database (MSigDB) v7.1 [29]. Then, we curated human gene sets from the collected items. A total of 2232 CP terms, including annotated gene sets from the BIOCARTA [30], Kyoto Encyclopedia of Genes and Genomes (KEGG) [31], Pathway Interaction Database (PID) [32], REACTOME, and Signaling Gateway databases [33], were curated. In addition, 5913 human GO terms, including biological process (BP), cellular component (CC), and molecular functions (MF), were obtained. We chose 123 inositol triphosphate receptor (ITPR) associated gene sets that contained ITPR genes. To avoid gene set enrichment toward overly broad terms, we filtered gene sets having more than 100 genes. We finally adopted 32 gene sets originating from 8 gene ontology terms, 4 KEGG pathways, 17 Reactome terms, and 3 Signaling Gateway terms. To estimate the activation level of gene sets in the expression data, we applied gene set variation analysis (GSVA) [34]. GSVA is a nonparametric and unsupervised gene set enrichment method that infers relative enrichment scores based on rank statistics of the expression values among all samples in a dataset. A GSVA enrichment score by sample matrix was generated for the target gene sets and used for further analysis using processed transcriptome data.

### 2.4. Survival Analysis

The survival rate and median survival time were estimated using the Kaplan-Meier (KM) method. Comparative survival analysis was performed between two groups: those with high- and low-score samples according to the expression level of the gene of interest or the gene set enrichment score. Iterative searches determined the thresholds for sample groups. We performed a log-rank test by setting an arbitrary threshold to between the 25th and 75th percentiles of gene expression or gene set enrichment score across datasets. The thresholds that had lowest *p*-value were chosen as the grouping threshold. In addition, hazard ratios (HRs) and 95% confidence intervals (95% CIs) were estimated using a univariate Cox regression model. The significance threshold was set to *p* < 0.05.

### 2.5. Cell Culture

Human non-small cell lung cancer cells (A549, H460, H1299, H358, H23) were kindly provided by Dr. Hae Yong Yoo (Samsung Medical Center, Sungkyunkwan University). A549 and H460 cells were cultured in Dulbecco’s Modified Eagle’s Medium (DMEM) and H1299, H358, and H23 cells were cultured in Roswell Park Memorial Institute (RPMI) 1640. These cells were supplemented with 10% fetal bovine serum (FBS), penicillin (100 units/mL), and streptomycin (100 μg/mL) in a humidified 5% CO_2_ incubator at 37 °C. Cell morphology was observed with an inverted microscope (Olympus IX81, Tokyo, Japan). All culture medium and supplements were obtained from Gibco (Grand Island, NY, USA).

### 2.6. Cell Proliferation Assay

Cell proliferation was determined using the 3-(4,5-dimethylthiazol-2-yl)-2,5-diphenyltetrazolium bromide (MTT, Sigma-Aldrich) assay. The cells were seeded in 96-well plates at a density of 4 × 10^3^ cells per well. After overnight incubation, the cells were treated with TFP and TFP analogs. At specific time points after incubation (24–72 h), MTT (1 mg/mL) was added to the medium for 2 h. The medium and MTT were removed, and then dimethyl sulfoxide (DMSO, AMRESCO, Solon, OH, USA) was added. We measured the absorbance at 570 nm with an Infinite M200 Pro (Tecan, Mannedorf, Switzerland). The IC_50_ values were calculated using GraphPad Prism (La Jolla, CA, USA). Briefly, the data were fit by non-linear regression, and IC_50_ values were determined using the equations Y = 100/(1 + 10^((LogIC50 − X)^
^∗ HillSlope)^), where Y is the percentage of cell viability relative to untreated samples.

### 2.7. Colony Formation Assay

The cells (5 × 10^3^) were plated in 6-well plates. The plates were incubated with TFP and 3dc, and the medium was changed every 3 days. After 8 days, the colonies were stained with crystal violet and counted. 

### 2.8. Soft Agar Colony Formation Assay

The cells (5 × 10^3^) were seeded in 0.35% agar medium with 10% FBS overlaid onto a previously prepared 0.7% base agar. The agar plates were incubated with TFP and 3dc. The medium was changed every 3 days. After 18 days, the colonies were stained with crystal violet and counted. 

### 2.9. Wound Healing Assay

The cells were plated and grown overnight to full confluence. A 200 μL pipette tip was used to create a scratch in the cell monolayer. The cells were washed with PBS and cultured in serum-free DMEM with TFP or 3dc. Then, the cells were subjected to the indicated treatment for 24 h and 48 h. The scratch gap width at each time point in each treatment group was measured and compared with the gap width at 0 h, which was arbitrarily set as 0. 

### 2.10. Western Blot Analysis

The cells were lysed in radioimmunoprecipitation assay (RIPA) buffer (Thermo Fisher Scientific, Rockford, IL, USA) containing 25 mM Tris-Cl (pH 7.6), 150 mM NaCl, 1% NP-40, 1% sodium deoxycholate, 0.1% SDS, protease inhibitor cocktail (Roche Life Science, Indianapolis, IN, USA), and phosphatase inhibitor cocktail (GenDEPOT, Barker, TX, USA). After brief sonication, the lysates were clarified by centrifugation at 12,000× *g* for 30 min at 4 °C. The protein concentrations were determined using a bicinchoninic acid (BCA) protein assay kit (Thermo Fisher Scientific). The proteins were loaded on 8–15% SDS-polyacrylamide gel for electrophoresis and then transferred to polyvinylidene difluoride membranes. The membranes were blocked with 5% skim milk in tris-buffered saline (TBS) containing 0.1% Tween-20 for 2 h at room temperature and then incubated with the appropriate primary and secondary antibodies. Labeled proteins were detected by chemiluminescence (ECL; Thermo Fisher Scientific) using a LAS 4000 system (Fujifilm, Tokyo, Japan). We used antibodies against p-ERK1/2 (Cell Signaling, Beverly, MA, USA, #9101), ERK1/2 (Cell Signaling, #9102), p-AKT (Cell Signaling, #9271), AKT (Cell Signaling, #9272), p-PI3K (Cell signaling, #4228), PI3K (Cell signaling. #4249), PARP (Cell Signaling, #9532), caspase 8 (Abcam, Cambridge, UK, ab25901), caspase 9 (Abcam, ab25758), and β-actin (Sigma, A2228).

### 2.11. Apoptosis and DNA Fragmentation Assay

Apoptosis was analyzed using the Annexin V-FITC Apoptosis detection kit (Thermo Fisher Scientific). The cells (2 × 10^6^) were plated in a 100-mm dish and treated with TFP and 3dc for 48 h. After treatment, the cells were harvested, washed with 1× PBS (plus 1% BSA, 2 mM EDTA), and stained with Annexin V-FITC and propidium iodide (PI). We then analyzed cells by flow cytometry using the Attune NxT flow cytometer (Thermo Fisher Scientific). DNA fragmentation was analyzed using the Cell Death Detection enzyme-linked immunosorbent assay (ELISA) kit (Roche Diagnostics, Mannheim, Germany).

### 2.12. Measurement of Changes in Intracellular Ca^2+^ Concentration

We measured changes in intracellular Ca^2+^ concentration ([Ca^2+^]_i_) as previously described, with slight modifications [35]. Briefly, A549 cells were incubated in a glass-bottom dish (SPL, Pocheon, Republic of Korea) with 5 μM Fluo 3-AM (Molecular Probes, Eugene, OR, USA), a Ca^2+^-sensitive fluorescent indicator, for 30 min at 37 °C. The samples were then washed three times with serum-free DMEM without phenol red. The dish was then loaded on a confocal microscope scanning system (IX70 Fluoview; Olympus) to record and analyze [Ca^2+^]_i_ changes. The fluorescence images were scanned every 5 sec using a 488 nm excitation argon laser and 530 nm long pass emission filters. The changes in [Ca^2+^]_i_ are represented as fluorescence intensity (FI, arbitrary units). The maximum and basal FI levels (F_max_ and F_base_) were analyzed with the highest Ca^2+^ peak shown after treatment with chemicals and with the FI level unchanged before treatment, respectively. Net changes in FI levels for each treatment were calculated with the formula (F_max_ − F_base_)/F_base_. To measure [Ca^2+^]_i_, a bath solution containing 125 mM NaCl, 5 mM KCl, 1 mM MgCl_2_, 5 mM glucose, 10 mM HEPES (with or without 1 mM CaCl_2_, pH 7.3), and 5 EGTA was used to create calcium-free conditions. All solutions were prepared with Milli-Q water (18.2 MΩ-cm at 25 °C).

### 2.13. Tumor Xenograft

A total of 30 female BALB/c nu/nu (athymic nude) mice were purchased from ORIENT BIO (Seongnam, Republic of Korea) and maintained on a 12 h light/12 h dark cycle, with food and water supplied ad libitum. For orthotopic tumor xenografts, A549 cells were intravenously injected into mice through the tail vein (2 × 10^6^ cells per mouse). After 3 days, the mice were randomly divided into three groups (*n* = 10 per group). TFP and 3dc were dissolved in DW and then diluted with saline (DW:saline = 1:9). These solutions were intraperitoneally injected at a dosage of 5 mg/kg/day, 5 days/week for 4 weeks. The treatment dose (5 mg/kg/day) was decided based on a previous report [13]. We measured body weight and monitored the health and behavior of the animals twice a week for 4 weeks. The control group received the vehicle alone. The animal experiment was terminated at 4 weeks after cell xenograft based on several references with similar protocol [36,37,38,39]. The animals were sacrificed with CO_2_ exposure (3L/min/cage), and tissues were dissected out. Humane endpoints criteria were specified as loss of body weight (more than 20%), eyes (narrow open or close, discharge), nose (discharge), activity, and posture (bent back, down head). We followed the Institutional Animal Care and Use Committee (IACUC) guidelines set by the Ministry of Agriculture, Food, and Rural Affairs and the Ministry of Food and Drug Safety of the Republic of Korea. All animal procedures followed the Animal Care and Use Guidelines of Gyeongsang National University (approval number: GNU-130813-M0055).

### 2.14. Tissue Preparation and Histological Analysis 

The mice were anesthetized and perfused through the left cardiac ventricle and ascending aorta with a fixative solution containing 4% paraformaldehyde. The lungs of mice were removed after perfusion and postfixed in the same fixative overnight at 4 °C. The tissues were treated with paraffin, sectioned to 5 μm thickness, stained with hematoxylin and eosin (H&E), and visualized with a light microscope (Olympus BX50).

### 2.15. Statistical Analysis

We performed statistical analysis using Student’s unpaired t-test and one-way analysis of variance (ANOVA) plus post hoc Dunnett’s test or Tukey’s test (GraphPad Prism). Data are shown as the mean ± standard error (SE). The intracellular Ca^2+^ concentration was analyzed with one-way ANOVA plus post hoc Tukey’s test (SPSS 18 software, SPSS Inc., Chicago, IL, USA). Electrophysiological data are represented as the mean ± standard deviation (SD), and *p* < 0.05 was considered significant. Transcriptome microarray analysis was conducted in the R statistical computing language, version 4.0.0 [40]. Data download and processing were performed using the R packages GEOquery [41] and affy [42]. The ComBat batch normalization algorithm was implemented with the sva R package. GSVA for gene set enrichment was performed using the R package GSVA [34]. Two-sample t-test and Benjamini-Hochberg adjustments for false discovery were conducted with a built-in R function (the stats package). Survival analysis, including KM curve and Cox regression analysis, was conducted with the survival [43] and survminer [44] packages. The data were plotted using the ggplot2 [45] and forestplot [46] R packages.

## 3. Results

### 3.1. TFP and TFP Analogs Inhibit the Proliferation of A549 Cells

We first screened several kinds of NSCLC cell lines (A549, H460, H1299, H358, H23) to evaluate the anti-proliferative effect of 3dc. The 3dc treatment significantly inhibited cell proliferation in several NSCLC cells despite genotype (p53 wild or mutant) or origin discrepancy (Appendix A). Then, we selected the A549 cell for further experiments because of its most common use in NSCLC cell lines. The A549 cell line is one of the most used xenograft lung cancer models and novel therapeutics due to the overexpression of HER-2 and EGFR receptors [47,48]. The structure of TFP and its analogs are presented in Figure 1A. To assess the effects of TFP and TFP analogs (3db, 3dc, 3dd, and 3fb) on A549 cell proliferation, MTT assay was performed and IC_50_s were calculated (Figure 1B). The cells were treated with TFP and TFP analogs at various concentrations (0, 1, 2, 5, 10, and 20 μM) for 48 h. We found that TFP and TFP analogs reduced cell proliferation. In particular, 3db, 3dc, 3dd, and 3fb inhibited cell proliferation more potently than TFP. Among 4 TFP analogs, 3dc was selected for further experiments. In addition, A549 cells were treated with TFP, 3dc, and cisplatin (Figure 1C), and 3dc treatment had a significantly greater inhibitory effect on cell proliferation than TFP or cisplatin. Cisplatin, a commonly used chemotherapeutic, was applied to verify the effects of TFP and 3dc. This result suggests that 3dc is more cytotoxic to A549 cells than TFP. 

### 3.2. TFP and 3dc Suppress the Anchorage-Dependent and Anchorage-Independent Growth and Migration of A549 Cells

To test the effects of TFP and 3dc on colony-forming ability, colony formation assay (Figure 2A) and soft agar colony formation assay (Figure 2B) were performed in A549 cells. TFP and 3dc (5 μM) treatment suppressed the anchorage-dependent and anchorage-independent colony formation of A549 cells. Furthermore, the colony-forming ability of 3dc-treated A549 cells was less than that of the control (CTL)- and TFP-treated cells, demonstrating that 3dc enhanced the inhibition of A549 cell growth.

Next, the effects of TFP and 3dc on A549 cell migration were examined with a wound healing assay. The cells were treated with TFP or 3dc (5 μM) for 24 h and 48 h. We found that TFP and 3dc treatment decreased cell migration (Figure 2C). In particular, 3dc treatment had a significantly more significant inhibitory effect on A549 cell migration than CTL and TFP.

### 3.3. 3dc Alters the Expression of Factors Related to Apoptosis and Survival

To determine the effects of TFP and 3dc treatment on apoptosis- and survival-related factors, we performed western blotting after treating cells with TFP or 3dc at 10 μM for 48 h (Figure 3A,B). Remarkably, 3dc treatment decreased p-ERK, p-AKT, and p-PI3K and increased the expression of cleaved-caspase 8 and cleaved-PARP. The activated form of caspase 9 looks increased by 3dc treatment, but the ratio of cleaved-caspase 9/pro-caspase 9 was not significantly changed. These data indicate that 3dc affects survival and apoptosis signaling pathways.

### 3.4. TFP and 3dc Induce A549 Cell Apoptosis and DNA Fragmentation

Previous research has shown that TFP induces apoptosis in human lung cancer cells [49,50]. Thus, we next investigated whether apoptosis contributes to TFP- and 3dc-related growth inhibition of A549 cells. Annexin V-FITC/PI staining showed that TFP or 3dc treatment increased the abundance of the apoptotic cell population (Figure 3C,D). Moreover, 3dc treatment induced DNA fragmentation (Figure 3E).

### 3.5. TFP and 3dc Treatment Increased Intracellular Ca^2+^ Levels

TFP and 3dc modulated the calcium homeostasis through ITPRs in glioma cells [21,51]. Thus, we aimed to evaluate the effect of TFP and 3dc on intracellular Ca^2+^ levels in A549 cells. The calcium ionophore A23187 (5 μM) was used as a positive control to increase intracellular Ca^2+^ levels in A549 cells. As expected, A23187 treatment transiently increased intracellular Ca^2+^ levels (Figure 4A). Treatment with TFP (10 μM) or 3dc (10 μM) markedly increased intracellular Ca^2+^ levels in A549 cells in the presence of extracellular Ca^2+^ (*n* = 150, TFP: 8.4 ± 2.5, 3dc: 8.0 ± 1.9). However, the Ca^2+^ peak significantly decreased by 72% in the absence of extracellular Ca^2+^ (*n* = 150, TFP: 2.4 ± 1.0, 3dc: 2.3 ± 0.8) (Figure 4B). There was no significant difference between TFP and 3dc (*p* > 0.05, Figure 4C). 

We found that TFP and 3dc produced characteristic Ca^2+^ responses (Figure 4B,C). Both TFP and 3dc induced long-lasting and high Ca^2+^ peaks, but the Ca^2+^ increase lasted over 1 min longer in the 3dc treatment group than in the TFP treatment group. In some cells, 3dc-induced Ca^2+^ level increases, and the level did not return to the basal level during the 15 min observation period (58%, 87/150).

In cells pretreated with thapsigargin (Thapsi), which stimulates Ca^2+^ release from intracellular Ca^2+^ stores, neither TFP nor 3dc increased intracellular Ca^2+^ levels (*n* = 20, Figure 4D). To determine the possible involvement of the phospholipase C (PLC)-ITP signaling pathway, we applied a specific PLC inhibitor (U73122, 5 μM) and an ITPR antagonist (xestospongin-C (Xes-C), 10 μM). In cells preincubated with U73122 or Xes-C for 10 min, TFP and 3dc did not increase intracellular Ca^2+^ levels (*n* = 90, Figure 4E). We found that U73122 and Xes-C treatment significantly reduced 3dc- and TFP-induced Ca^2+^ increases (*p* < 0.05, Figure 4F). However, in some cells, Xes-C treatment did not completely block the effects of TFP. In addition, 3dc treatment slightly decreased basal Ca^2+^ levels in cells pretreated with U73122 and Xes-C. Moreover, 3dc treatment induced bleb formation (Figure 4G).

### 3.6. ITPR Expression and Associated Pathway Activity Are Related to Patient Survival Probability

We hypothesized that the molecular pattern of ITPRs is related to the clinical benefit of lung cancer patients. To investigate the connection of ITPRs and associated pathway activity with clinical outcome, we conducted a survival analysis using integrated public transcriptome datasets labeled with survival time. We identified differences in survival outcomes depending on variations in ITPR expression and related pathway activation scores. The log-rank test of the KM curve and the HR from the Cox proportional hazard model were used to estimate the survival probability.

We first assessed associations between the expression values of all ITPRs (ITPR1, ITPR2, ITPR3) and overall survival in lung cancer patients. For the survival analysis, we divided patients into high- and low-expression groups using the gene expression values of interest with the best thresholds (see Materials and Methods). Patients with increased ITPR1 expression showed a significantly higher survival rate than those with increased expression of other ITPRs (Figure 5A,B, log-rank *p*-value = 0.0011, HR = 0.60, 95% CI 0.44–0.82). The ITPR3 expression level was also significantly associated with the overall survival rate (Figure 5A,B, P = 0.016, HR = 0.63, 95% CI 0.42–0.92). To confirm that a significant change in survival rate did not appear by chance, we performed a similar analysis for single datasets before integration. We obtained an equivalent tendency of association between ITPR expression and survival outcome in these studies as well (Figure 5C).

We further conducted a similar analysis of the activity of ITPR-associated functional groups. We chose 32 ITPR-associated gene sets that contained ITPR genes selected from MSigDB [29] gene sets (see Materials and Methods). We conducted survival analysis of patients stratified into high- and low-enrichment score groups and identified 11 functional gene sets that were significantly related to overall survival (Table 1). 

### 3.7. Inhibitory Effects of TFP and 3dc on Metastasis in A549 Cell-Derived Xenograft Model

In vitro studies revealed that TFP and 3dc treatment inhibited the proliferation, migration, and growth of A549 cells. Therefore, we also used in vivo studies to investigate the effects of TFP and 3dc on metastasis. To create a tumor xenograft model, A549 cells were injected into the tail veins of nude mice. The mice were then treated with vehicle, TFP, or 3dc (5 mg/kg/day, intraperitoneally, 5 days/week for 4 weeks). The bodyweight of mice did not change during the experiment (data not shown). Remarkably, TFP and 3dc treatment suppressed lung metastasis. To confirm this result, the lung tissues were stained with H&E (Figure 6A). Indeed, TFP and 3dc treatments suppressed the formation of metastatic lung nodules (Figure 6B). 

## 4. Discussion

New drug development requires major investments in terms of capital and time. Drug development costs can be decreased by repurposing previously developed drugs with known safety profiles. This approach may be convenient for identifying new cancer therapeutics, especially given the emerging need to overcome drug resistance in cancer treatment. Currently used cancer drugs also have side effects, such as nausea, vomiting, hair loss, and pain, causing great suffering for many patients during cancer treatment. Chemotherapy is a common approach to treating lung cancer, the leading cause of cancer-related mortality in the world. In NSCLC, chemotherapy may be used in different situations, including before surgery to shrink a tumor, after surgery to kill cancer cells that may be left behind, as primary treatment for advanced cancers, or in patients who are not healthy enough for surgery. In addition, chemotherapy outcomes are dependent on the NSCLC stage. The chemotherapy drugs most often used for NSCLC are cisplatin, carboplatin, paclitaxel, albumin-bound paclitaxel, docetaxel, gemcitabine, vinorelbine, irinotecan, etoposide, vinblastine, and pemetrexed [1]. 

Interestingly, some antipsychotic drugs have been shown to possess anticancer activity. Phenothiazines are relatively stable and widely used antipsychotic drugs that mainly act on central dopamine receptors. Chlorpromazine, a representative drug of the phenothiazine family, has been shown to enhance the cytotoxic effects of tamoxifen in breast cancer [52]. Other reports have shown that chlorpromazine can specifically inhibit mitotic kinesin KSP/Eg5 to cause mitotic arrest, further inhibiting tumor cell proliferation [53], and that it can selectively exert cytotoxic effects on lymphoblastoid tumor, neuroblastoma, NSCLC, and breast cancer cells while sparing normal cells [54]. Phenothiazines also enhance the sensitivity to cisplatin [55] and reverse tumor resistance to other chemotherapeutic drugs [56]. Combining chlorpromazine and TFP at standard clinical doses promoted apoptosis in leukemia and lymphoma without affecting normal cells [57]. Together, this evidence strongly indicates that phenothiazines can inhibit tumor proliferation. Previous studies have found that phenothiazines can inhibit tumor proliferation [58]. Additionally, because phenothiazines have antiemetic, sedative, and analgesic effects, these drugs could reduce the side effects of chemotherapy and alleviate anxiety, insomnia, and other psychological symptoms relevant to cancer treatment. Chlorpromazine was discovered in 1952 and has broad antitumor application prospects because of its widespread use and specific side effects.

This study found that the phenothiazine derivative TFP and its analog 3dc reduced the proliferation of A549 human lung cancer cells. TFP is a long-established, broadly used conventional antipsychotic drug that has been demonstrated to be effective at low doses and considered safe since the 1960s [59]. As antipsychotics, TFP has a high affinity for D1 and D2 receptors and is also known as a CaM antagonist. Several studies have shown the effect of Ca^2+^/CaM on cellular signaling [60,61,62,63]. Ca^2+^/CaM signaling promotes tumor cell viability and motility by activating the AKT signaling pathway [64,65,66]. 

Previous studies reported that antipsychotic agents have an antiproliferative effect [55]. Our results showed that 3dc treatment inhibited A549 cell proliferation more potently than TFP treatment and that TFP inhibited A549 cell proliferation, migration, and tumor formation. In addition, TFP is known to inhibit cancer stem cell growth and overcome lung cancer drug resistance [67]. However, TFP still has side effects. A previous study of mice with orthotopic brain tumors [21] found that TFP treatment induced parkinsonian-like behavior and extrapyramidal side effects. We decided to synthesize TFP analogs to reduce side effects and increase the anticancer efficacy. More than 60 TFP analogs were synthesized and their activity was screened with intracellular calcium level monitoring. In our previous report, 3dc has the most intracellular calcium increasing activity [21]. Among four analogs that have higher antiproliferative activity than TFP, 3dc was selected for further experiments. Although 3dc is not the most potent IC_50_ compared to the other three analogs, it shows stable dose sensitivity. No sign of extrapyramidal side effects was detected after injecting 3dc. Thus, 3dc, a TFP analog that is predicted to have few side effects and high efficacy, may be a candidate treatment.

The anticancer effects of TFP may be related to Ca^2+^/CaM signaling and AKT activation. In a previous study, we reported that TFP treatment induced an intracellular Ca^2+^ increase in U87MG glioma cells [51]. These increases were mediated by the opening of ITPR1 and ITPR2 on the endoplasmic reticulum membrane (ER). Intracellular Ca^2+^ concentration plays a pivotal role in cell activities, including the cell cycle, proliferation, and apoptosis. The major signaling pathway involved in Ca^2+^ release from the ER is PLC-ITPR [68]. Moreover, ER Ca^2+^-homeostasis was altered and ITPR expression was increased in NSCLC cells [69]. In the present study, TFP or 3dc treatment induced intracellular Ca^2+^ release in A549 lung cancer cells. These increases were significantly blocked by the inhibition of PLC or ITPR. In particular, 3dc treatment induced bleb formation in A549 cells, suggesting apoptotic progression. The 3dc-induced apoptosis was confirmed by Annexin V-FITC/PI staining and DNA fragmentation assay. 

AKT is phosphorylated and activated by Ca^2+^/CaM-dependent protein kinase via a PI3K-dependent pathway. These molecular events lead to BAD phosphorylation, and the cells are protected from serum withdrawal-induced apoptosis [70]. PI3K is regulated by phosphatase and tensin homolog, also known as TGFβ-regulated and epithelial-cell-enriched phosphatase, or mutated in multiple advanced cancers [71,72,73,74]. Our results showed a decrease in proliferation and a reduction in the number and size of tumor spheres with 3dc treatment. This observation may be explained the downregulation of AKT phosphorylation caused by 3dc’s effects on CaM. This model suggests that TFP inhibits AKT phosphorylation by antagonizing Ca^2+^/CaM signaling, and this is supported by other findings that treatment with a Ca^2+^ channel inhibitor or CaM antagonists decreases the basal level of AKT phosphorylation [74]. In the present study, 3dc treatment significantly decreased the expression of p-PI3K and downstream p-AKT.

The present study revealed that 3dc significantly induced the apoptosis of A549 cells via Annexin V-FITC/PI staining and DNA fragmentation. In addition, increased expressions of cleaved-caspase 8 and cleaved-PARP confirmed 3dc-induced apoptosis. 

Ca^2+^ influx promotes tumor cell migration and metastasis [75,76], and CaM antagonists inhibit tumor cell invasion and metastasis [77,78,79]. Because of the vital importance of Ca^2+^/CaM in the regulation of cell proliferation and apoptosis, CaM antagonists have been widely used in the clinic as anticancer agents [80]. Notably, the activation gene sets related to Ca^2+^ -channel mediated signaling pathways and cytosolic Ca^2+^-involved pathways showed a significant association with overall survival. This result indicates that the activation of ITPRs and relevant functional pathways is related to survival outcomes in lung cancer patients. In support of the experimental evidence from the cell model, ITPRs—specifically, ITPR1 overexpression and associated pathway activation—were significantly related to the clinical outcome of patients. This supports that there is a crucial mechanism by which TFP and 3dc act through ITPR that can be applied for lung cancer therapy in real-world patients.

The in vivo portion of this study revealed that 3dc strongly suppressed A549 cell lung metastasis in an orthotopic tumor xenograft model. As seen in tissue sections from three experimental groups, both TFP and 3dc treatment suppressed lung metastasis. Notably, tissues from the 3dc-treated group were similar to the normal tissues. These in vivo results are consistent with other studies [35,81]. Although this study screened TFP analogs and identified 3dc as a candidate anti-lung-cancer therapeutic, its toxicity must be evaluated by clinical trials. Prior to this study, we administered 3dc to mice in the tumor xenograft model, and no behavioral changes were observed [21]. In terms of TFP dose in the tumor xenograft model, we referred to several previous studies for TFP dose in the animal model. Most researchers mainly used 5 mg/kg/day [67], 10 mg TFP/kg/day [82,83], or even 20 mg/TFP/kg [84] in orthotopic liver [83] or lung [67] xenograft models. In the present study, we used 5 mg/TFP or 3dc/kg/day for in vivo experiment. This concentration is approximately 24 mg/day/60 kg. The practical dose of TFP for patients with mental disorders is usually 10 mg/day/60 kg. A dose of 100 to 150 mg/day is also used in a particular set of patients [85]. We believe that this dose is applicable to cancer patients with severe clinical status.

In conclusion, although it has been accepted that several kinds of antipsychotics have potential anticancer activity, the clinical use of antipsychotics for cancer treatment needs further evaluation and improvement. We showed that a synthetic analog of TFP has intense anti-lung-cancer activity. Thus, antipsychotics can prevent the progression of NSCLC and support the development of new anticancer drugs in a drug repositioning manner.

## Figures and Tables

**Figure 1 biomedicines-10-01046-f001:**
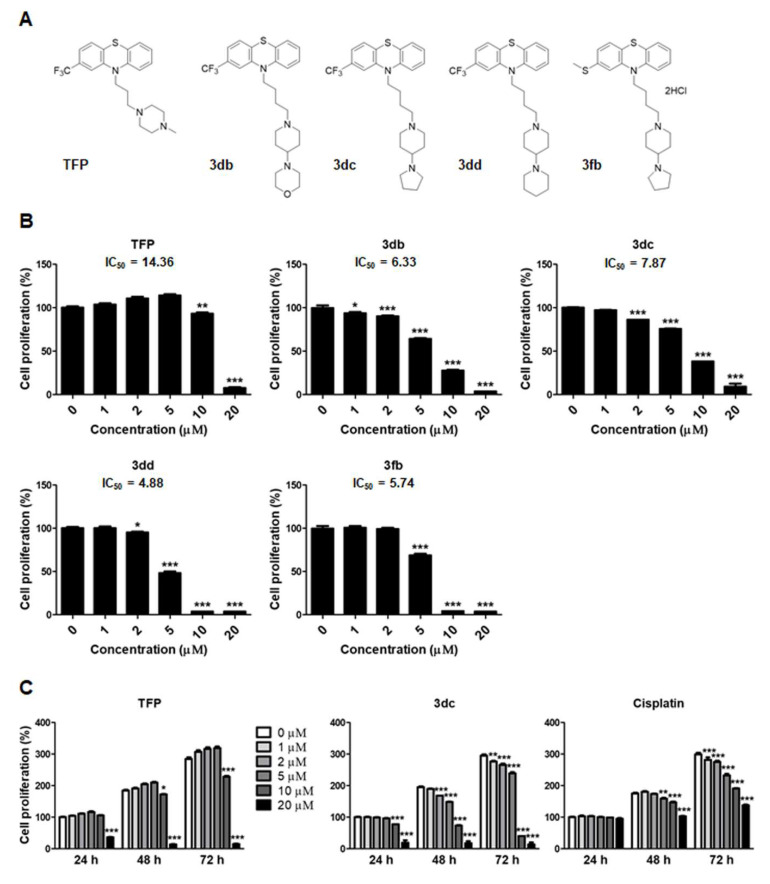
Treatment with TFP and 3dc inhibited A549 cell proliferation. (**A**) Structure of TFP and its analogs. (**B**) A549 cells were treated with TFP and TFP analogs (3db, 3dc, 3dd, and 3fb) for 48 h. Cell proliferation was assessed using the MTT assay. (**C**) A549 cells were treated with TFP (left), 3dc (center), and cisplatin (right) at various concentrations (1, 2, 5, 10, and 20 μM) for 24–72 h. CTL (0 μM for 24 h). The data are shown as the mean ± SE. * *p* < 0.05, ** *p* < 0.01, *** *p* < 0.001 vs. CTL (0 μM).

**Figure 2 biomedicines-10-01046-f002:**
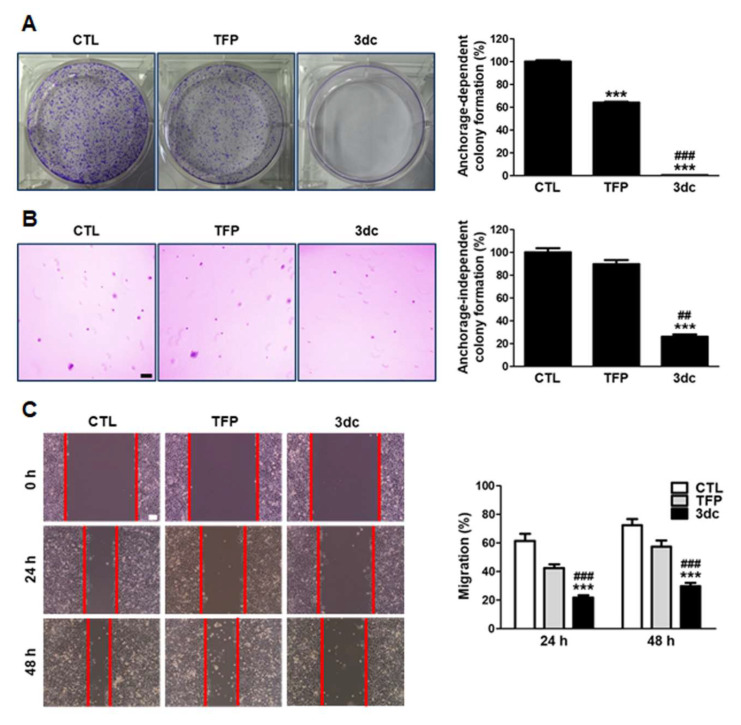
Treatment with TFP and 3dc suppressed anchorage-dependent and anchorage-independent A549 cell growth. Cells were seeded and treated with 5 μM of TFP and 3dc. (**A**) After 8 days of incubation, colonies were photographed with a digital camera and manually counted. (**B**) After 18 days of incubation, colonies were photographed at a magnification of 40X (scale bar represents 200 μm). (**C**) TFP and 3dc decreased A549 cell migration. Cell growth surface was scratched and treated with 5 μM of TFP and 3dc. After 24 h and 48 h of incubation, migrating cells were photographed at a magnification of 100× (scale bar represents 100 μm). Lines indicate wound boundaries after scratch. The data are shown as the mean ± SE. *** *p* < 0.001 vs. CTL, ^##^
*p* < 0.01, ^###^
*p* < 0.001 vs. TFP.

**Figure 3 biomedicines-10-01046-f003:**
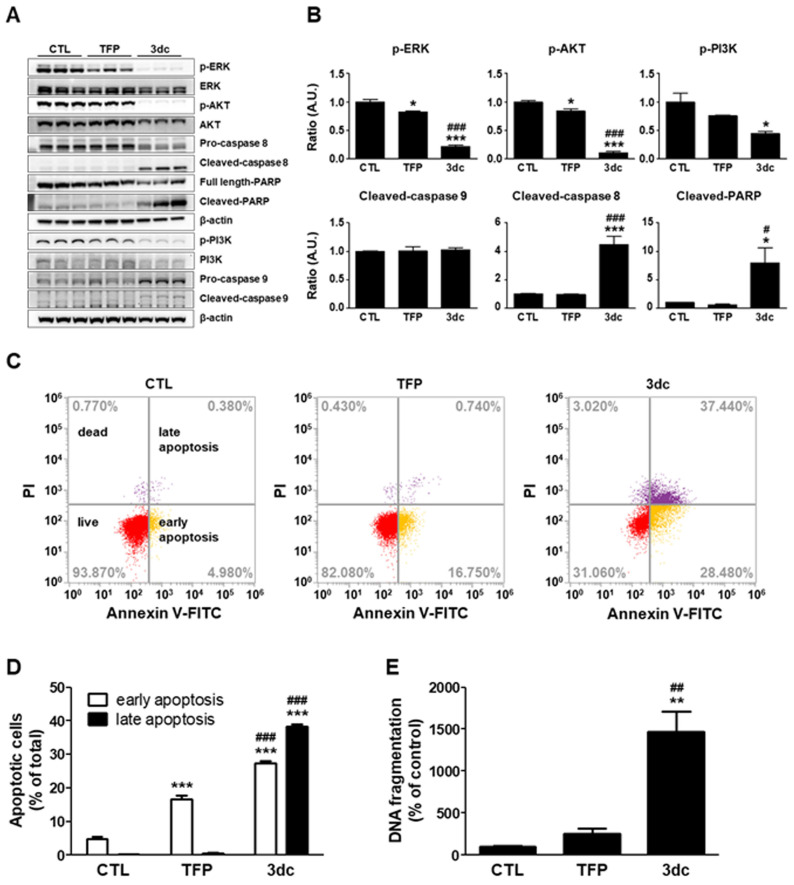
Western blot analysis of apoptosis- and survival-related factors. (**A**) A549 cells were treated with 10 μM TFP and 3dc for 48 h. Whole cell lysates were analyzed by western blot with antibodies specific to p-ERK, ERK, p-AKT, AKT, p-PI3K, PI3K, caspase 8, caspase 9, and PARP. β-actin was used as a protein loading control. (**B**) Protein expression levels were assessed using ImageJ software and represented as the change of each group relative to the CTL group using arbitrary units. (**C**) Flow cytometric analysis of TFP- and 3dc-treated cells. A549 cells were treated with 10 μM TFP and 3dc for 48 h and then stained with Annexin V-FITC and PI. CTL (left), TFP (center), 3dc (right). (**D**) Percentage of apoptotic cells. (**E**) Quantification of DNA fragmentation after 48 h incubation of A549 cells with 10 μM TFP and 3dc. The data are shown as the mean ± SE. * *p* < 0.05, ** *p* < 0.01, *** *p* < 0.001 vs. CTL, ^#^
*p* < 0.05, ^##^
*p* < 0.01, ^###^
*p* < 0.001 vs. TFP.

**Figure 4 biomedicines-10-01046-f004:**
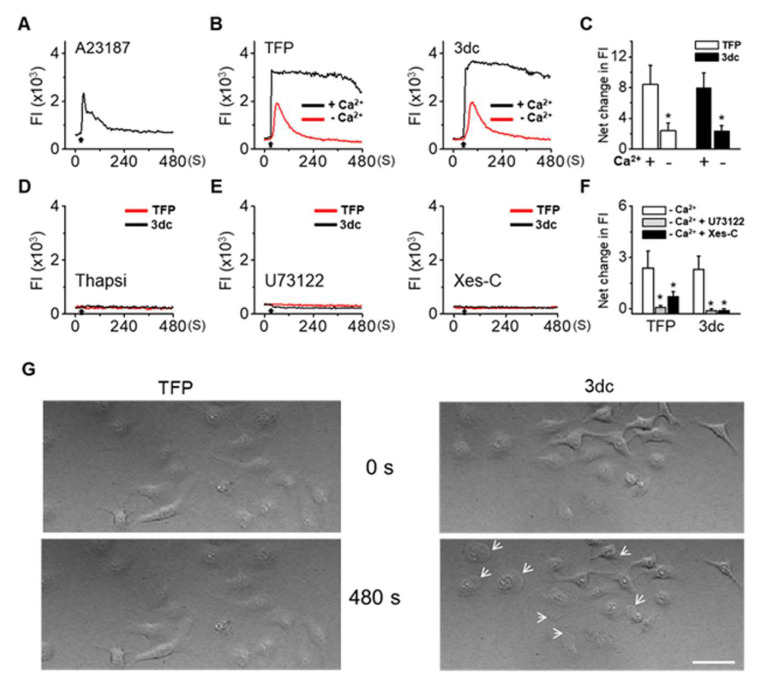
TFP and 3dc treatment increased intracellular Ca^2+^ levels. (**A**) The calcium ionophore A23187 increased intracellular Ca^2+^ levels in A549 cells. (**B**) Typical Ca^2+^ waves in response to TFP (left) and 3dc (right) treatments. A549 cells were treated with 10 μM TFP or 10 μM 3dc in the presence and absence of extracellular Ca^2+^. Calcium-free conditions were produced with 0 mM CaCl_2_ and 5 mM EGTA. (**C**) Summary of data in panel B. + and – represent the presence and absence of Ca^2+^, respectively. Each bar is the mean ± SD obtained from three independent experiments. * *p* < 0.05 compared to the presence of Ca^2+^. (**D**) There was no response to TFP and 3dc in the cells pretreated with 1 μM thapsigargin for 30 min in the absence of extracellular Ca^2+^. (**E**) Involvement of the PLC and ITPR pathways in the TFP- and 3dc-induced Ca^2+^ increase. Cells were pretreated with U73122 (5 μM, left) and Xes-C (10 μM, right) prior to TFP or 3dc application. (**F**) Summary of the data in panel E. Each bar is the mean ± SD obtained from three independent experiments. * *p* < 0.05 compared to the absence of Ca^2+^. (**G**) 3dc-induced bleb formation in A549 cells. The images were captured 8 min after treatment with 3dc. The scale bar represents 50 μm. White arrows indicate bleb formation. FI: Fluorescence intensity (arbitrary units) of cells. Black arrows represent the addition of chemicals.

**Figure 5 biomedicines-10-01046-f005:**
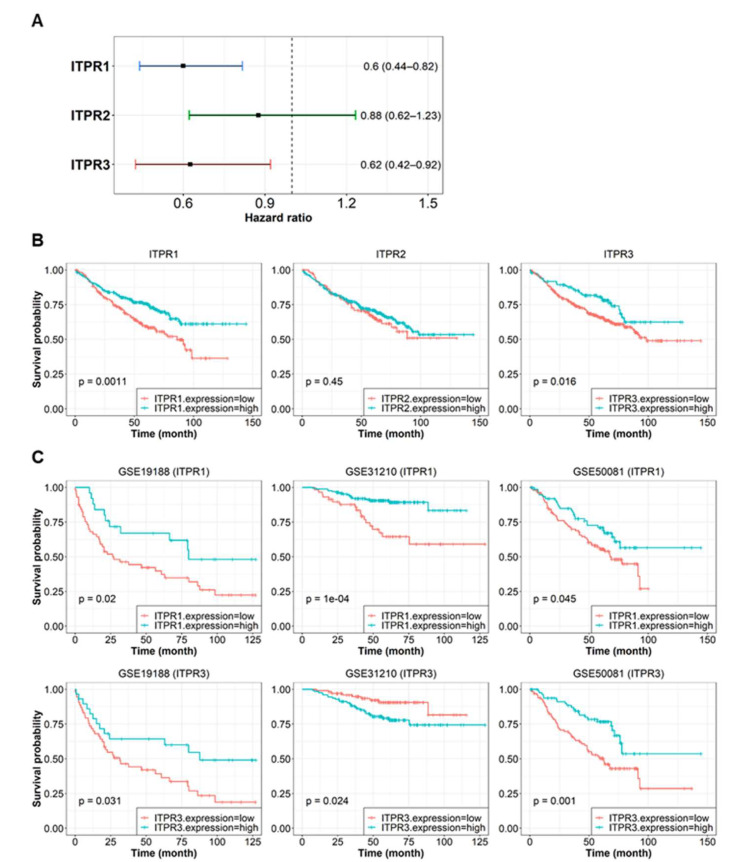
Correlations between the expression of ITPRs (ITPR1, ITPR2, ITPR3) and the clinical outcome of lung cancer patients. The expression values from combined lung cancer patient transcriptome data (GSE19188, GSE31210, GSE50081) were analyzed. (**A**) HR between the high- and low-expression level groups for each ITPR. The reference group was set to the group with a low expression level. The 95% CIs are shown with error bars. (**B**) KM curve of two groups divided by ITPR1 (left), ITPR2 (center), and ITPR3 (right) expression levels. The *p*-values of the stratified log-rank test between the two groups are described. (**C**) KM curve analysis in separated microarray datasets. The survival rate of groups stratified according to ITPR1 (upper) or ITPR3 (lower) expression levels were assessed in the individual datasets (GSE19188, GSE31210, GSE50081). HRs and log-rank *p*-values are denoted on a logarithmic scale.

**Figure 6 biomedicines-10-01046-f006:**
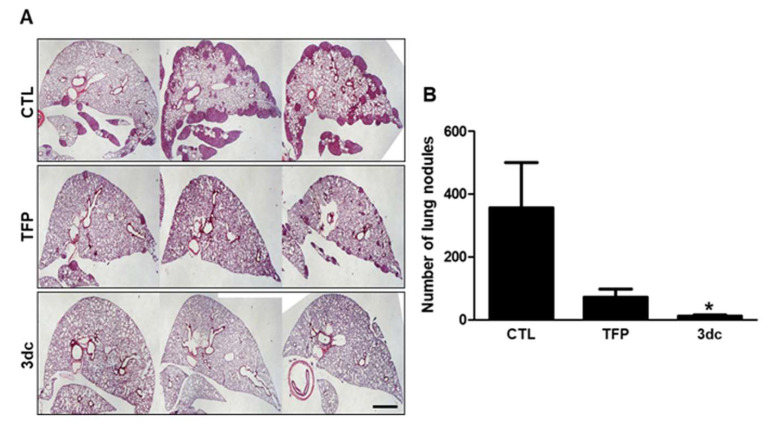
Inhibitory effect of TFP and 3dc on metastasis in A549 cell-derived xenograft model. A549 cells were intravenously injected into mice through the tail vein. After 3 days, the mice were divided into three groups (CTL, TFP, and 3dc; *n* = 10 in each group). The mice were then injected intraperitoneally with CTL (vehicle), TFP, and 3dc (5 mg/kg/day, 5 days/week, for 4 weeks). (**A**) Lungs from the mice in the orthotopic xenograft model were harvested, sliced, and H&E stained. Scale bar represents 1 mm. (**B**) The graph shows the number of lung nodules. The values are mean ± SE. * *p* < 0.05 vs. CTL.

**Table 1 biomedicines-10-01046-t001:** Functional gene sets that are significantly related to lung cancer patients’ survival outcomes. Survival analysis was conducted between high- and low-enrichment score sample groups. HR and 95% CI values were calculated from the Cox proportional hazard model, and the *p*-value of the log-rank test comparing two groups was calculated from the KM curve. The significance threshold was set to *p* < 0.05.

Functional Gene Set	Log-Rank *p*-Value	HR	95% CI
SIG_PIP3_SIGNALING_IN_B_LYMPHOCYTES	2.30 × 10^−5^	0.517	0.38–0.71
REACTOME_EFFECTS_OF_PIP2_HYDROLYSIS	7.13 × 10^−4^	0.588	0.43–0.8
REACTOME_DAG_AND_IP3_SIGNALING	2.63 × 10^−3^	0.624	0.46–0.85
REACTOME_ION_HOMEOSTASIS	7.99 × 10^−3^	0.638	0.46–0.89
GO_LIGAND_GATED_CALCIUM_CHANNEL_ACTIVITY	0.0127	0.672	0.49–0.92
REACTOME_ANTIGEN_ACTIVATES_ B_CELL_RECEPTOR_BCR_LEADING_TO_ GENERATION_OF_SECOND_MESSENGERS	0.0145	0.667	0.48–0.92
GO_REGULATION_OF_CARDIAC_CONDUCTION	0.0149	0.669	0.48–0.93
KEGG_PHOSPHATIDYLINOSITOL_SIGNALING_SYSTEM	0.0226	0.639	0.43–0.94
REACTOME_REGULATION_OF_INSULIN_SECRETION	0.0257	0.690	0.5–0.96
REACTOME_G_PROTEIN_MEDIATED_EVENTS	0.0372	0.719	0.53–0.98
GO_CALCIUM_ION_IMPORT_INTO_CYTOSOL	0.0391	0.664	0.45–0.98

## Data Availability

The datasets used and/or analyzed during the current study are available from the corresponding author on reasonable request.

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
