# Peer review of "Trifluoperazine and Its Analog Suppressed the Tumorigenicity of Non-Small Cell Lung Cancer Cell; Applicability of Antipsychotic Drugs to Lung Cancer Treatment"

_biomedicines, 2022, doi:10.3390/biomedicines10051046_

Round 1

Reviewer 1 Report

The paper presented by the authors has three different parts:

  • Statistics of lung cancer in psychotic populations that pretends to show that lung cancer has a lower incidence in these populations.
  • In vitro experimental evidence showing that trifluoperazine has anti-cancer effects.
  • In vivo experimental evidence showing lack of metastasis in the treated animals.

1.- Statistics

The statistical data presented in the paper is strongly biased because:

  1. The fact that a patients is psychotic does not mean that he was treated with trifluoperazine.
  2. The most statistically relevant study they present in the paper is that of Goldacre (17) which concludes that “We found no evidence that schizophrenia confers protection that schizophrenia confers protection against cancer in general
  3. The paper by Grinshpoon (5) reaches exactly the opposite result of what the authors support: “Lung cancer was significantly higher in men born in Asia–Africa diagnosed with schizophrenia than in the respective comparison population group” . In addition, this paper cannot be used as a proof because the population is exclusively Jewish, which means that there is a racial bias from the beginning. To be remembered the higher BRCA1/2 mutation in Jewish populations.
  4. The work by Raviv (26) is limited to the analysis of prostate cancer: “ The present study suggests a reduced rate of prostate cancer in patients admitted for schizophrenia. There are several possible explanations for this finding including chronic state of hyperprolactinemiainduced by antipsychotic drugs.”  Interestingly, this study finds exactly the opposite of what the authors sustain: the risk of lung cancer in schizophrenia is 1.43. However, the authors do not use this data but that of prostate.
  5. The study by Dalton (19) gave neutral results: “The standardized incidence ratio(SIR) of lung cancer was marginally reduced”
  6. The study by Osborn (25) also arrived to different conclusion than those of the authors: In a cohort analysis within a large UK primary care database, the incidence of colo-rectal, breast and lung cancer, and of all common cancers, did not differ significantly in people with SMI, including schizophrenia, compared with people without SMI. Our results do not support enhanced screening procedures for cancer in people with SMI.
  7. The paper by Hippisley-Cox (20) also concludes that “The increased risk of colon cancer is particularly marked in patients with schizophrenia who take antipsychotic medications.”

First conclusion: Five of the 10 papers included in references show a completely different conclusion from what the authors pretend to support. Suggestion: delete all the statistical paragraph.

2.- In vitro experiments

The concentrations used by the authors in their experiments are more than a 1,000 fold higher than what can be achieved in a patient. This kind of experiments are an exercise in futility.

Second conclusion: None of the experiment were performed at clinically achievable concentrations. The authors should clear this point.

3.- In vivo experiments

In this case there is a clear difference favoring trifluperazine and its derivatives as an anti-metastatic drug. However, this should not be considered  as synonym of tumorigenicity suppression as the authors pretend. There are many factors involved in metastasis beyond tumor initiation: niche, circulation, metastogenes, etc.

Author Response

  1. Statistics

The statistical data presented in the paper is strongly biased because:

  1. The fact that a patients is psychotic does not mean that he was treated with trifluoperazine.
  2. The most statistically relevant study they present in the paper is that of Goldacre (17) which concludes that “We found no evidence that schizophrenia confers protection that schizophrenia confers protection against cancer in general
  3. The paper by Grinshpoon (5) reaches exactly the opposite result of what the authors support: “Lung cancer was significantly higher in men born in Asia–Africa diagnosed with schizophrenia than in the respective comparison population group” . In addition, this paper cannot be used as a proof because the population is exclusively Jewish, which means that there is a racial bias from the beginning. To be remembered the higher BRCA1/2 mutation in Jewish populations.
  1. The work by Raviv (26) is limited to the analysis of prostate cancer: “ The present study suggests a reduced rate of prostate cancer in patients admitted for schizophrenia. There are several possible explanations for this finding including chronic state of hyperprolactinemiainduced by antipsychotic drugs.”  Interestingly, this study finds exactly the opposite of what the authors sustain: the risk of lung cancer in schizophrenia is 1.43. However, the authors do not use this data but that of prostate.
  2. The study by Dalton (19) gave neutral results: “The standardized incidence ratio(SIR) of lung cancer was marginally reduced”
  3. The study by Osborn (25) also arrived to different conclusion than those of the authors: In a cohort analysis within a large UK primary care database, the incidence of colo-rectal, breast and lung cancer, and of all common cancers, did not differ significantly in people with SMI, including schizophrenia, compared with people without SMI. Our results do not support enhanced screening procedures for cancer in people with SMI.
  1. The paper by Hippisley-Cox (20) also concludes that “The increased risk of colon cancer is particularly marked in patients with schizophrenia who take antipsychotic medications.”

First conclusion: Five of the 10 papers included in references show a completely different conclusion from what the authors pretend to support. Suggestion: delete all the statistical paragraph.

Response: Thank you for your constructive suggestions. We recognized your comment that people with schizophrenia do not fully represent people treated with antipsychotics, including TFP. We reviewed cohort studies that we introduced to our meta-analysis, and none of them included information about antipsychotics prescription, especially TFP. We agree that our statistical meta-analysis of the cohort studies is insufficient to support our conclusion that TFP has anticancer effects at the patient level. Therefore, we decided to exclude our cohort meta-analysis parts from Materials and Methods and Results sections according to your suggestion.

However, we still believe that it should not be ruled out that the factors of schizophrenic patients, including antipsychotic medication, can be effective against lung cancer. Because numerous cohort studies (70% of target studies) report the decreased risk of lung cancer in schizophrenic cohorts in their statistics, our meta-analysis's summarized statistic showed decreased incidence ratio with a notable confidence interval. As you pointed out in the comments and first conclusion, individual studies show different conclusions about the standardized incidence ratio of lung cancer in schizophrenic patients. This was also mentioned in Discussion section of our manuscript (line 510-512). In this situation in previous studies, we applied a meta-analysis technique to assess summarized effect across heterogeneous studies with increased statistical power and precision. During meta-analysis procedure, effect size of each study is weighted according to estimated sample variance from confidence interval, which is calculated using sample size in their original studies. Through this process, we conducted a meta-analysis considering each study's sample size and effect size. As a result, we were able to derive a trend in which the incidence of lung cancer decreased in patients with schizophrenia. In addition, various psychotropic drugs were reported to show anti-tumor activity in cellular experimental models (mentioned in line 61, line 499-502). We interpreted that these facts emphasize the need to analyze the anticancer effects caused by antipsychotic medication.

Taking this into account, we added statements to the introduction that argue the necessity for further study on the anticancer effect of antipsychotics medication based on the clues from previous experimental and epidemiological studies. We hope that this revised part of the introduction will strengthen the relevance of our research design.

We have revised Abstract as follows:

We analyzed publicly available clinical data to evaluate the possible association between schizophrenia and lung cancer risk. Then, wWe examined the effects of trifluoperazine (TFP), a commonly used antipsychotic drug, and its synthetic analogs on A549 human lung cancer cells as well as primary lung cancer cells from patients.

We have revised Abstract as follows:

The clinical data meta-analysis revealed decreased lung cancer incidence in schizophrenic patients, suggesting that antipsychotics have an anticancer effect and k Key genes and mechanisms possibly affected by TFP and 3dc are significantly related to better survival outcomes in lung cancer patients.

We have revised Introduction as follows:

Recently, antipsychotics, antidepressants, and neuroleptics, have shown remarkable antiproliferative activity [3]. Moreover, low prevalence of various cancer types was observed in patients with schizophrenia [4,5]. In the case of lung cancer, several cohort studies have reported cases of reduced incidence ratio in the schizophrenic population [6-13]. This implies that some factors of schizophrenic patients including neuroleptic treatment might have an anticancer effect. However, the patient-level analysis related to the treatment of specific antipsychotics is not sufficiently investigated. Based on these biological and epidemiological grounds, it is necessary to further analyze the anticancer effect of antipsychotics.

We have revised Introduction as follows:

The present study aimed to evaluate the possibility of antipsychotics as anticancer therapeutics for NSCLC using publically available clinical data.

We have revised Materials and Methods as follows:

Transcriptome microarray analysis and meta-analysis were was conducted in the R statistical computing language, version 4.0.0 [40].

We have revised Discussion as follows:

Analysis of publicly available clinical data demonstrated the possibility that antipsychotics can prevent the progression of NSCLC. Furthermore, We showed that a synthetic analog of TFP has intense anti-lung cancer activity.

We have deleted Section 2.2 in Materials and Methods, Section 3.1 in Results, line 508-516 in Discussion, Figure 1, Figure S1, Figure S2 and Table S1.

  1. In vitro experiments

The concentrations used by the authors in their experiments are more than a 1,000 fold higher than what can be achieved in a patient. This kind of experiments are an exercise in futility. Second conclusion: None of the experiment were performed at clinically achievable concentrations. The authors should clear this point.

Response: As you commented it, the concentration of drugs used in vitro experiments is not applicable to patient directly. Before animal experiment or human application, researchers perform an in vitro experiment to screen the candidates. In many other reports, IC50 of TFP ranged 7 - 16 mM. In the present study, IC50 of TFP and 3dc were 13.36 mM and 7.87 mM, respectively. As you commented on PDF file, blood concentration of human after TFP administration was quite lower than the dose of in vitro experiment. However, direct comparison with cell line experiments and clinical application has considerable discrepancies. We believe animal experiment is more profitable to calculate the clinical application.

Related references:

  1. Chen, Q.Y.; Wu, L.J.; Wu, Y.Q.; Lu, G.H.; Jiang, Z.Y.; Zhan, J.W.; Jie, Y.; Zhou, J.Y. Molecular mechanism of trifluoperazine induces apoptosis in human A549 lung adenocarcinoma cell lines. Mol. Med. Rep. 2009, 2, 811-817.
  2. Yeh, C.T.; Wu, A.T.; Chang, P.M.; Chen, K.Y.; Yang, C.N.; Yang, S.C.; Ho, C.C.; Chen, C.C.; Kuo, Y.L.; Lee, P.Y.; et al. Trifluoperazine, an antipsychotic agent, inhibits cancer stem cell growth and overcomes drug resistance of lung cancer. Am. J. Respir. Crit. Care Med. 2012, 186, 1180-1188.

Xia, Y., Jia, C., Xue, Q., Jiang, J., Xie, Y., Wang, R., Ran, Z., Xu, F., Zhang, Y., Ye, T.  Antipsychotic Drug Trifluoperazine Suppresses Colorectal Cancer by Inducing G0/G1 Arrest and Apoptosis. Front. Pharmacol. 2019, 10, 1029.

Qian, K., Sun, L., Zhou, G., Ge, H., Meng, Y., Li, J., Li, X., Fang, X. Trifluoperazine as an alternative strategy for the inhibition of tumor growth of colorectal cancer. J. Cell. Biochem. 2019, 120, 15756-15765.

Wang, B., Zhou, W., Wang, Y., Li, R. Trifluoperazine Inhibits Mesangial Cell Proliferation by Arresting Cell Cycle-Dependent Mechanisms. Med. Sci. Monit. 2017, 23, 3461-3469.

  1. In vivo experiments

In this case there is a clear difference favoring trifluperazine and its derivatives as an anti-metastatic drug. However, this should not be considered as synonym of tumorigenicity suppression as the authors pretend. There are many factors involved in metastasis beyond tumor initiation: niche, circulation, metastogenes, etc.

Response: Thank you for your kind and critical comment. As you commented it, our in vivo experiment was performed to analyze the metastasis. We changed our description of results section (3.7). Please see the revised manuscript.

  1. Comment on PDF file

The number of cases of each publication should be mentioned. Finally, it is important to know the total number of patients involved. Are they discussing large or small populations?

This is particularly important because three of the ten studies show a very increased hazard ratio: Goldacre, Grinshpoon, and Raviv. Osborn and Dalton are non significant. Furthermore, Goldacre is very significant in the proportion and shows an increased hazard ratio.

To the best of my criteria, I think this statistic is biased.

Response: Please see the response about Statistics.

Reviewer 2 Report

In their manuscript, JY Jeong et al evaluate the anti-cancer effects of the anti-psychotic drug trifluoperazine (TFP) and of one of its analogs. Drug repurposing is a good strategy for fast development of novel therapeutic strategies, and in this context, antipsychotics have been already shown to elicit anti-cancer effects and need further studies to clearly assess their potential. In this study the authors clearly show the anti-proliferative effects of TFP and its analog 3dc. The work presented is interesting and well performed. 

I only have few comments about the presented data and the conclusion drawn.

1) Its not clear enough why the authors employed TFP analogs and what advantages these analogs may have over the original drug. I suggest the authors comment more on this aspect. Also, why the authors have chosen 3dc over the other analogs? Please explain better. Also please comment more on the possible "reduced" side effects of 3dc over TFP.

2)  Figure 5. The authors claim that TFP induces bleb formation. I believe that more in depth analyses are needed to support this hypothesis. The images presented are not explicative enough. Can the authors run IF analyses (with specific markers) on treated A549 cells to visualize blebs?

3) Authors chose to investigate the role of ITPR pathway. It is not clear how they were prompted to test the importance of this pathway. The authors have addressed in other tumor types the role of these molecules, but they should clarify why they chose it in this study. I believe that the data presented in figure 5 are not sufficient to justify the correlation analyses on ITPR. Please explain better

4) Figure 7: The tail injection of tumor cells can not be presented as an "orthotropic" tumor xenograft model, but it represents an in vivo induced metastasis assay. Please correct the text accordingly. Also, how the authors can dissect whether the effect of TFP and 3dc they see is due changes in the proliferative capacity of cells, homing potential, resistance in the circulation? A standard subcutaneous injection would work best to assess the anti-proliferative effects of TFP and 3dc on A549 cells in vivo. Anyhow the authors can not conclude that their data demonstrate that "TFP and 3dc suppress tumorigenicity and metastasis".

5) please correct typos: e.g.: line 429 "For the survival of analysis".

Reviewer 3 Report

The current manuscript is a exact replica of the work reported in PMID: 29614416, except that they are using different cell line. From my point of view the rational is still weak the repurposing idea is better. The introduction is weak. I do not say that the manuscript has its value I am saying that this  is a repetition of something that they already did. Based also in anecdotical data not strong base. I believe that the article is publishable but not in biomedicines to keep its high impact.

Author Response

The current manuscript is a exact replica of the work reported in PMID: 29614416, except that they are using different cell line. From my point of view the rational is still weak the repurposing idea is better. The introduction is weak. I do not say that the manuscript has its value I am saying that this is a repetition of something that they already did. Based also in anecdotical data not strong base. I believe that the article is publishable but not in biomedicines to keep its high impact.

Response: Thank you for your critical comments. When our research team published a report about the anticancer effect of caffeine and glioma (Kang et al., 2010, Cancer Research), Richard Eric Kast (Department of Psychiatry, University of Vermont, Burlington, VT) suggested the original idea TFP might have anticancer effect on glioma. Then, we performed experiment to evaluate the antiglioma activity of TFP. As previously reported (Kang et al., 2017, Mol. Cancer Ther.), TFP showed powerful antiglioma activity in vitro but not in vivo. Thus, we try to synthesize TFP analogs for a better outcome and possible clinical application. However, the efficacy was only merely improved (Kang et al., 2018, Eur. J. Med. Chem.). For the screening purpose, several kinds of cancer cell lines were tested. TFP and its analog 3dc showed a strong antiproliferative effect on lung cancer cells. First question was whether antipsychotics lower the lung cancer incidence. We applied both bioinformatics and experiments. As you commented, it was not easy to use precise clinical data with personal medications. However, we thought our present manuscript is not merely a replica of the previous report. Please see the details within our manuscript.

Reviewer 4 Report

In the manuscript "Trifluoperazine and its analog suppressed the tumorigenicity of non-small cell lung cancer cell; Applicability of antipsychotic drugs to lung cancer treatment"

-the introduction is exhaustive and pertinent to the subject dealt with.

-materials and methods: are clearly described and suitables to the type of experiments carried out.

-results: in my opinion the authors in this manuscript have applied a correct experimental design, from the use of data present in appropriate databases, to cellular studies to clarify the effectiveness of the compounds, their molecular targets, the cellular mechanisms with which they carry out their antitumor activity and finally in vivo studies on mice.

-discussion: the authors explain in a rational and concrete way the results obtained and the conclusions are in line with the results obtained.

-references: authors indicate appropriate references

Author Response

Comments and Suggestions for Authors

In the manuscript "Trifluoperazine and its analog suppressed the tumorigenicity of non-small cell lung cancer cell; Applicability of antipsychotic drugs to lung cancer treatment"

-the introduction is exhaustive and pertinent to the subject dealt with.

-materials and methods: are clearly described and suitables to the type of experiments carried out.

-results: in my opinion the authors in this manuscript have applied a correct experimental design, from the use of data present in appropriate databases, to cellular studies to clarify the effectiveness of the compounds, their molecular targets, the cellular mechanisms with which they carry out their antitumor activity and finally in vivo studies on mice.

-discussion: the authors explain in a rational and concrete way the results obtained and the conclusions are in line with the results obtained.

-references: authors indicate appropriate references

Response: Thank you for your kind and favorable comments.

Round 2

Reviewer 2 Report

The authors have addressed all my comments.

Reviewer 3 Report

The authors addressed my concerns and the introduction seems much better. I am good believer of the repurposing which the authors highlighted.